# Low-temperature paddlewheel effect in glassy solid electrolytes

Jeffrey G. Smith [1,2] & Donald J. Siegel [1,2,3,4,5 ✉]

Glasses are promising electrolytes for use in solid-state batteries. Nevertheless, due to their amorphous structure, the mechanisms that underlie their ionic conductivity remain poorly understood. Here, ab initio molecular dynamics is used to characterize migration processes in the prototype glass, $75Li_2S–25P_2S_5$. Lithium migration occurs via a mechanism that combines concerted motion of lithium ions with large, quasi-permanent reorientations of $PS_4^{3-}$ anions. This latter effect, known as the 'paddlewheel' mechanism, is typically observed in high-temperature crystalline polymorphs. In contrast to the behavior of crystalline materials, in the glass paddlewheel dynamics contribute to Lithium-ion mobility at room temperature. Paddlewheel contributions are confirmed by characterizing spatial, temporal, vibrational, and energetic correlations with Lithium motion. Furthermore, the dynamics in the glass differ from those in the stable crystalline analogue, $γ-Li_3PS_4$, where anion reorientations are negligible and ion mobility is reduced. These data imply that glasses containing complex anions, and in which covalent network formation is minimized, may exhibit paddlewheel dynamics at low temperature. Consequently, these systems may be fertile ground in the search for new solid electrolytes.

[1] Mechanical Engineering Department, University of Michigan, Ann Arbor, MI 48109, USA. [2] Joint Center for Energy Storage Research, University of Michigan, Ann Arbor, MI 48109, USA. [3] Materials Science & Engineering, University of Michigan, Ann Arbor, MI 48109, USA. [4] Applied Physics Program, University of Michigan, Ann Arbor, MI 48109, USA. [5] University of Michigan Energy Institute, University of Michigan, Ann Arbor, MI 48109, USA. ✉email: djsiege@umich.edu

Replacing the liquid electrolytes used in conventional Li-ion batteries with a solid electrolyte (SE) would improve battery safety and potentially allow the use of higher-capacity (metallic) negative electrodes[1,2]. Of the small but growing number of potential SEs, the lithium thiophosphate family has emerged as a promising candidate system due to its high ionic conductivity and favorable formability[3–7].

Studies of ionic conductivity in thiophosphates were first reported in 1980, when it was observed that substituting sulfur for oxygen in phosphates increased the conductivity from 1 to 3 orders of magnitude[8,9]. More recently, the thiophosphate with composition $Li_{9.54}Si_{1.74}P_{1.44}S_{11.7}Cl_{0.3}$ has demonstrated the highest room-temperature conductivity ($25 \, mS \, cm^{-1}$) of any SE[5]. High Li-ion conductivity has also been reported in several related phases, such as $Li_7P_3S_{11}$ ($17 \, mS \, cm^{-1}$)[3] and $Li_{10}GeP_2S_{12}$ ($12 \, mS \, cm^{-1}$)[4]. Notably, all of these systems are derived from (or related to) the parent $Li_2S–P_2S_5$ (LPS) glass[7]. Thus, understanding the phenomena responsible for fast ion conduction in LPS glass could allow its favorable properties to be generalized to other systems, aiding in the design of broader categories of SEs[6,7].

Glasses lack long-range atomic order. Moreover, the energy landscape for ion migration in these systems can be influenced by subtle variations in local coordination, and has been proposed to be dynamically heterogeneous, i.e., consisting at any instant in time of quiescent domains that describe the majority of the system, with a few "active" regions comprising the remainder[10–12]. Simulations of model glasses find that the active regions can exhibit cooperative (or correlated) atomic motion[13–22], while motion in the quiescent domains is negligible.

Correlation effects in ionic motion were first discussed by Bardeen and Herring[23]. Correlations can exist between successive jumps of an ion. Furthermore, these jumps can also influence the migration behavior of neighboring ions. In the absence of these correlations, the self-diffusion, $D$, and conductivity, $\sigma$, are related by the Nernst–Einstein equation: $D = \sigma k_B T / nq^2$. (Here, $k_B$ is Boltzmann's constant, $T$ is the absolute temperature, $n$ is the number density of mobile species, and $q$ is the effective charge.) Early measurements on SEs found evidence for deviations from Nernst–Einstein behavior, suggesting the presence of correlated transport mechanisms[24–30].

Due to their high concentration of mobile carriers, modern SEs may also exhibit correlated transport[31,32]. Indeed, recent classical and AIMD models of crystalline Li-ion SEs have proposed that cooperative motion underlies fast ion conduction[33–37]. For example, the difference in conductivity observed between cubic and tetragonal $Li_7La_3Zr_2O_{12}$ has been rationalized by the presence of concerted motion in the highly conductive cubic phase, and the absence of this phenomena in the nonconductive tetragonal polymorph[33–35]. Similarly, Xu et al.[36] observed "string-like" lithium migration in crystalline $Li_{10}GeP_2S_{12}$, while He et al.[37] argued that cooperative phenomenon was the "origin of fast ion diffusion in superionic conductors." This latter study found that migration energy barriers in several crystalline SEs were more consistent with experimental measurements when accounting for multi-ion migration, rather than assuming single-ion hopping alone.

A second, but arguably less well-known category of correlation effects links ionic mobility with the rotational properties of complex (polyhedral) anions. Nearly a century ago, Benrath and Drekopf observed an anomalous rise in the ionic conductivity of lithium sulfate, $Li_2SO_4$, coinciding with the transformation of the low-temperature monoclinic structure to the face-centered cubic α phase at 580 °C[38,39]. This observation went mostly unnoticed until 1957, when Førland and Krogh-Moe[40] studied the structure of the high-temperature α phase using X-ray diffraction. They remarked that some interatomic distances in their proposed

structure were implausible, unless rotations or oscillations of the tetrahedral $SO_4^{2-}$ anions were allowed[40]. A connection between these two observations in α-$Li_2SO_4$—high conductivity and rotational disordering of the anions—was proposed in 1972 in the form of a transport model referred to as the "cog-wheel" mechanism[41]. In this model, the tetrahedral sulfate ions can rotate, and in so doing are proposed to open lower-energy migration passageways between lithium sites, thereby facilitating the migration of Li ions. The high diffusivity observed for large tracer cations and anions in α-$Li_2SO_4$ has been cited as support for this mechanism[41].

The dynamic, rotationally disordered behavior of the sulfate anions in α-$Li_2SO_4$ was later confirmed by neutron diffraction. These measurements suggested that oxygen from the $SO_4^{2-}$ tetrahedra was spherically disordered around the central sulfur[42]. Similarly, single-crystal diffraction subsequently revealed that Li was evenly distributed in a spherical shell surrounding the anions[43]. These observations were consistent with the larger unit cell volume in α-$Li_2SO_4$ compared with its low-temperature analog, with the additional volume believed to facilitate anion rotations[44]. Börjesson and Torell[45] measured the activation energies for anion rotation and lithium diffusion in α-$Li_2SO_4$, and found that they were very similar, 0.40 and 0.34 eV, respectively. Taken together, these studies imply a strong dynamic coupling between the rotational motion of the $SO_4^{2-}$ anions and the translational motion of the Li cations, i.e., a "paddlewheel" mechanism.

Similar cation–anion coupling has been observed in other high-temperature sulfate phases that exhibit fast ion conduction, such as $LiNaSO_4$ and $LiAgSO_4$[46], and in the high-temperature modification of sodium ortho-phosphate, $Na_3PO_4$[47,48]. In the opposite sense, limited rotational motion of $SO_4^{2-}$ anions in $Na_2SO_4$ at elevated temperatures has been correlated with low Na-ion conductivity[45]. More recently, in borohydrides and closo-boranes (containing anions such as $BH_4$, $B_{10}H_{10}$, $B_{12}H_{12}$, etc.), the observation of enhanced conductivity that coincides with a transformation to the high-temperature structure has been attributed to the rotational properties of the anions[49–57]. Similarly, in Ba-doped $LaGaO_4$, oxide-ion conduction at elevated temperatures has been proposed to occur through a cooperative transport process involving the rotation of $GaO_4$ tetrahedra[58].

The preceding examples illustrate that the rotational dynamics of anions can significantly enhance cation mobility. It is therefore reasonable to hypothesize that the paddlewheel effect could be exploited to design new SEs with high ionic conductivity[59,60]. Unfortunately, exploiting these rotational degrees of freedom can present a challenge, as the paddlewheel mechanism is typically observed in high-temperature polymorphs having expanded volumes[39–43,45–54,61]. These polymorphs can be difficult to stabilize at ambient temperatures, as is desired for the operation of a battery[51,62]. The closo-carborane with composition $NaCB_9H_{10}$ represents one recent exception, in that it exhibits an order–disorder-phase transformation for anion disordering at 297 K, and a corresponding Na-ion conductivity of $0.03 \, S \, cm^{-1}$[63]. This compound was identified following a series of studies on other closo-boranes and -carboranes, all of which exhibit transition temperatures above ambient temperature[50,55–57]. Recent studies on $Li_{10}GeP_2S_{12}$, $Na_{11}Sn_2PS_{12}$, glass–ceramics, and other systems have also alluded to contributions to ionic conductivity from anion rotations[64–69]. In polymeric conductors, segmental motion of the polymer chains is believed to contribute to ion migration[70].

Here, we report an observation of the paddlewheel effect at ambient temperature in the Li-ion conducting glass, $75Li_2S–25P_2S_5$. More specifically, ab initio molecular dynamics (AIMD) simulations were used to reveal the static and dynamic

atomic-scale features that underlie ionic diffusivity in glassy $Li_3PS_4$. A model of the amorphous LPS structure was generated using melt-and-quench AIMD. The static structure was characterized using the partial pair distribution function (p-PDF), the coordination number $n(r)$, and total neutron PDF. The calculated structure is shown to closely match the measured neutron PDF, and the atom pairs responsible for each peak in the measured PDF are identified. Clear peaks are present in the Li–(P,S) pair distribution, indicating the presence of local order associated with neighboring lithium and P/S anions. In contrast, the Li–Li pair distribution exhibits no pronounced peaks, suggesting that Li is disordered. Moreover, the coordination number of Li does not exhibit a plateau-like region. The absence of a plateau suggests that Li ions experience a range of solvation environments, with coordination numbers ranging from 3 to 5.

Lithium migration is observed to occur via a complex mechanism that combines concerted motion of lithium ions with large, quasi-permanent rotations of the $PS_4^{3-}$ tetrahedra. Paddlewheel contributions are confirmed through analyses of spatial, temporal, vibrational, and energetic correlations with Li motion. Li-ion migration events at 300 K coincide with the reorientation of coordinating $PS_4^{3-}$ tetrahedra—a direct observation of the paddlewheel effect. In addition, the velocity autocorrelation power spectrum shows a strong overlap between Li vibrational modes and anion rotations/librations, while the activation energies for Li migration are similar to those for anion rotation. The dynamics in the glass are shown to differ from those in the crystalline ($\gamma$-$Li_3PS_4$) analog, where contributions from anion reorientations are negligible.

The presence of the paddlewheel effect in LPS glass at low temperatures is straightforward to explain. First, this glass contains $PS_4$ complex anions; rotations of these anions will exert a force on the cations. Second, due to imperfect ionic packing (amorphous structure), the glass has a lower density than the crystalline analog, $\gamma$-$Li_3PS_4$. This lower density provides the additional free volume needed to allow for anion rotations. Third, while a lower density (relative to the crystalline phase) is a property common to any glass, a distinguishing feature of the $75Li_2S$–$25P_2S_5$ composition is the relative absence of a long-ranged covalent network comprising longer-chain $P_xS_y$ anions. The presence of such a network can impede $PS_4$ rotations and slow down Li migration[71]. Notably, the 75–25 composition maximizes the number of compact and rotatable $PS_4$ anions, while minimizing less rotationally active longer-chain $P_xS_y$ components[71]. Glasses that exhibit this combination of features—that contain complex anions, yet have limited network-forming ability—have the potential to exhibit paddlewheel dynamics at low temperature, and thus may be promising SEs.

## Results

**Static structure.** Figure 1 shows the structure of glassy $Li_3PS_4$ (LPS) at $T = 300$ K and $P = 1$ bar, as generated by ab initio MD. (Additional details describing the structure generation procedure are provided in the "Methods" section and in Supplementary Fig. 1.) The resulting density was 1.56 g cm$^{-3}$; for comparison, the calculated density of crystalline $\gamma$-$Li_3PS_4$[72] (the stable crystalline phase at ambient conditions) is 1.83 g cm$^{-3}$, while the measured densities for the glass when prepared at room temperature vary from 1.45 to 1.68 g cm$^{-3}$, depending on the applied pressure[73]. The instantaneous and average densities of the glass as a function of pressure are shown in Supplementary Fig. 2 and Supplementary Table 1. $PS_4$ tetrahedra comprise the anionic components of the computed glass structure at 300 K; at this temperature, no other complex anion types (such as $P_2S_6$ and $P_2S_7$) were observed[7].

The static structure of the computed LPS model was characterized using three techniques: (i) the partial p-PDF, $g_{\alpha\beta}(r)$, (ii) the coordination number, $n_{\alpha\beta}(r)$, and (iii) the total neutron-weighted PDF, $G'(r)$. The p-PDF is a commonly used measure of the local structure of glasses and liquids. It is defined as[74]

$$g_{\alpha\beta}(r) = \frac{1}{\rho_\beta N_\alpha} \left\langle \sum_{\alpha,i} \sum_{\beta,j\neq i} \delta(r - r_{ij}) \right\rangle. \quad (1)$$

Here, $\delta$ is the Dirac delta function, $N_\alpha$ is the number of species of type $\alpha$, and $\rho_\beta$ is the number density of species $\beta$. The summations run over all atoms $i$ and $j$ of all types $\alpha$ and $\beta$. $r_{ij}$ is the scalar distance from ion $i$ to $j$, and the angled brackets represent a time average. The coordination number, $n_{\alpha\beta}(r)$, of an ion of type $\alpha$ is the average number of ions of type $\beta$ within a distance $r$ of $\alpha$. It is defined as an integral of the p-PDF: $n_{\alpha\beta}(r) = 4\pi \int \rho_\beta g_{\alpha\beta} r^2 dr$[74]. Finally, the total neutron-weighted PDF, $G'(r)$, was calculated to allow comparisons with recent neutron measurements of the structure of glassy LPS[7]. $G'(r)$ is given by

$$G'(r) = \left( \sum_\alpha^n c_\alpha \bar{b}_\alpha \right)^{-2} \sum_{\alpha,\beta}^n c_\alpha c_\beta \bar{b}_\alpha \bar{b}_\beta g_{\alpha\beta}(r). \quad (2)$$

Here, $c_\alpha$ and $\bar{b}_\alpha$ are the concentration and coherent-bound neutron-scattering length of species of type $\alpha$, respectively[75,76].

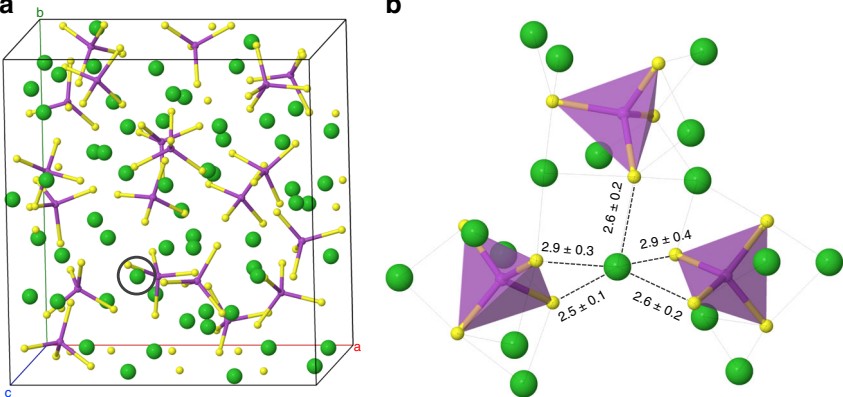

**Fig. 1 Computed structure of glassy $Li_3PS_4$ at 300 K and 1 bar generated from melt-and-quench ab initio MD. Green spheres represent lithium ions, phosphorus is magenta, and sulfur is yellow. a** Computational cell. A representative Li ion is identified with a black circle. The local solvation shell of this ion is magnified in (**b**), where numbers indicate the mean and standard deviation of nearest-neighbor sulfur distances in Å, averaged over 10 ps of MD.

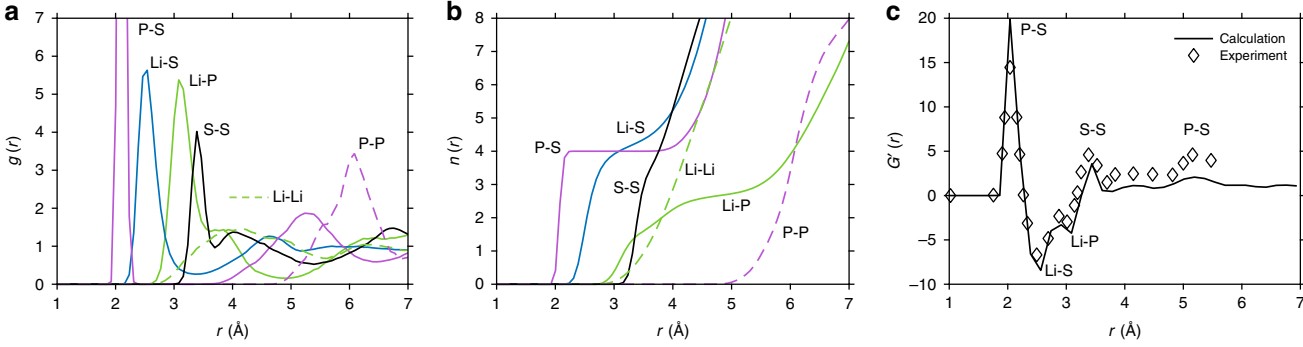

**Fig. 2 Characterization of the static structure of glassy Li₃PS₄ at ambient temperature. a** Partial pair distribution function, $g(r)$. **b** Coordination number, $n(r)$. **c** Total neutron-weighted pair distribution function, $G'(r)$, (solid line) compared with experimental neutron diffraction data (diamonds) reported by Ohara et al.[7]. Atom pair distances associated with the peaks in $G'(r)$ are labeled.

Figure 2a shows p-PDFs for all relevant atom pairs in glassy LPS at 300 K. As expected, the most prominent peak in the p-PDF is due to the P–S covalent bond found in the $PS_4^{3-}$ tetrahedra, with a maximum at 2.1 Å. The local environment of lithium ions is shown by the blue (Li–S), green (Li–P), and dashed green (Li–Li) lines. Clear peaks exist in the Li p-PDF at 2.5 and 3.1 Å, indicating the presence of local ordering of Li with respect to S and P, respectively. Nevertheless, both the Li–S and Li–P PDFs are nonzero at distances slightly larger than their peak values. (For example, the Li–P PDF exhibits a shoulder at 3.8 Å.) This behavior differs from the localized peak associated with covalent P–S bonds, and indicates greater variability in the local structure of Li. In contrast, the Li–Li PDF exhibits no sharp peaks, suggesting that the Li distribution is disordered. Longer-ranged features in the pair distances include those for S–S and P–P. S–S distances are shown by the black line; peaks at 3.4 Å and 4.0 Å correspond to intra- and inter-$PS_4^{3-}$ distances, respectively. P–P distances (dashed magenta line) exhibit a peak at 6.1 Å, corresponding to the average spacing of $PS_4^{3-}$ centers.

Figure 2b illustrates the average coordination number of ions of type $\alpha$ by ions of type $\beta$ (labeled as "$\alpha$–$\beta$" in the figure) as a function of distance, $r$. Turning first to the coordination of P by S, $n(r)$ displays a clear plateau for $r = 2$–4 Å. This behavior reflects the well-defined P–S covalent bonds present in the $PS_4^{3-}$ tetrahedra. (Beyond 4 Å, $n(r)$ increases continuously due to coordination of P by S from different tetrahedra.) In contrast to the behavior of P–S, the coordination number of Li does not exhibit a plateau-like shape. The absence of a plateau region suggests that Li ions experience a range of different coordination environments in LPS glass. The coordination number of the first coordination shell is defined as the integral of the PDF from 0 to $r_{max}$, where $r_{max}$ corresponds to the first minimum in the PDF. For Li–S pairs, this minimum occurs at 3.3 Å, corresponding to a coordination of 4.25 sulfur atoms. The fact that $n(r)$ is a non-integer further supports the notion of variable coordination of Li. Finally, a combined analysis of the Li–S $g(r)$ and $n(r)$ suggests that Li ions experience coordination environments generally ranging from 3 to 5 nearest-neighbor sulfur anions.

Recently, Ohara et al.[7] performed time-of-flight neutron diffraction measurements on LPS glasses having the same composition as those modeled here. Those measurements suggested that the glass structure consisted almost entirely of *ortho*-thiophosphate anions, consistent with the present computational findings. Figure 2c compares the measured and computed total neutron-weighted PDFs, $G'(r)$. Very good agreement is obtained: all major peaks in $G'(r)$ are reproduced by both theory and experiments. (This agreement is also maintained in the computational models having larger densities that were generated

using higher pressures, Supplementary Fig. 3a.) Furthermore, since Eq. (2) is the weighted sum of calculated partial PDFs (Eq. (1)), the individual pair distances responsible for each peak in $G'(r)$ can be predicted, and are shown in Supplementary Fig. 3b. Visible in both calculation and experiment are (i) P–S bonds within and between $PS_4^{3-}$ tetrahedra at 2.1 and 5.2 Å, respectively; (ii) S–S pairs at 3.4 Å arising from intra-$PS_4^{3-}$ interactions; (iii) the Li-anion peaks with characteristic distances of approximately 2.5 Å for Li–S and 3.1 Å for Li–P. All of these distances are very similar to those reported in the p-PDFs of Fig. 2a. Finally, the simulated $G'(r)$ data also explain why peaks associated with Li–Li and P–P correlations are not observed in the experimental measurements. The absence of the former peaks can be explained by the disordered distribution of Li (see the dashed green line in Fig. 2a), while the relatively low concentration of P explains the latter omission. In total, the good agreement with the neutron diffraction data suggests that the computational model accurately reproduces the short-ranged order present in the glass.

**Dynamics**. Li migration events were characterized using a protocol previously used to analyze dynamics in LLZO and in glass-forming liquids[35,77]. Following those studies, we define the functional $h_i(t; a, \Delta t, t_a)$, which identifies long-lived Li-ion displacements of at least a distance $a$ occurring at time $t$:

$$h_i(t; a, \Delta t, t_a) = \prod_{t' = \frac{t_a}{2} - \Delta t}^{t_a/2} \theta(|r_i(t + t') - r_i(t - t')| - a). \quad (3)$$

$h_i = 1$ for atoms that undergo such a displacement; otherwise $h_i = 0$. Here, $\theta(x)$ is the Heaviside step function, while the difference $r_i(t + t') - r_i(t - t')$ represents the displacement of atom $i$. The displacement threshold, $a$, was set to 1.6 Å, in accordance with ref. [35]. Supplementary Figs. 4 and 5 show the effect of varying the size of this threshold. $\Delta t = 3$ ps, is a residence time that both precedes and follows a displacement event, and ensures that these displacements are long lived. Finally, $t_a = 9$ ps, is a longer time window that includes the residence and the transition times. Eq. (3) implies that a migration event will occur over a time that is at most $t_a - 2\Delta t$, which, with the current parameters, is equal to 3 ps.

Figure 3a plots $h_i$ vs. simulation time for each of the 60 Li ions contained in the computational cell during the first of 6 independent MD runs performed at 300 K (data from the other runs are described below). The data show that a total of seven ions underwent long-lived migration events. Summing Eq. (3) at each time $t$ allows for the identification of ions whose migration events occur simultaneously (or near simultaneously). These events are shown in Fig. 3b. The first event observed occurs over

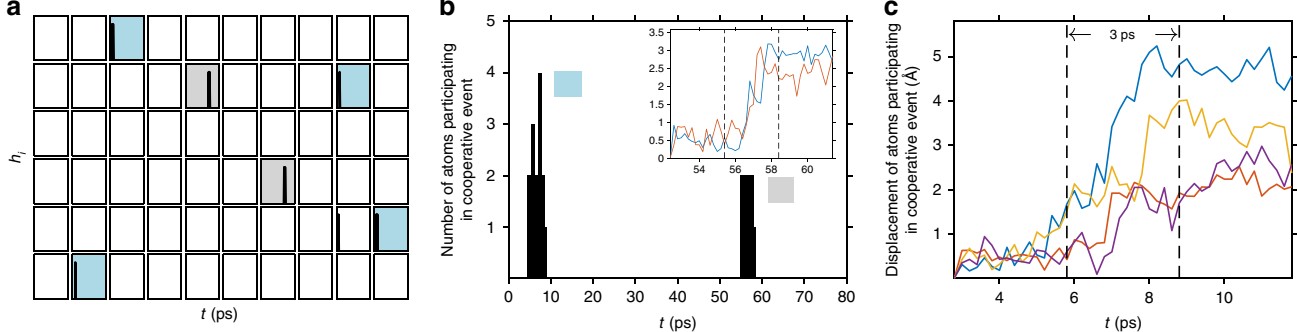

**Fig. 3 Detection of Li-ion migration events. a** Identification of specific Li ions that migrate at 300 K. Each rectangle individually plots $h_i$ (Eq. 3) vs. simulation time for the 60 Li ions in the simulation cell. **b** Number of atoms participating in a cooperative migration event, and the time at which those events were observed. Blue and gray shading in panels (**a**), (**b**) identify, respectively, the Li ions that migrate during the first and second cooperative events shown in panel (**b**). **c** Displacements of cooperatively migrating Li ions participating in the first cooperative event (identified with blue shading in (**a**)). The inset in (**b**) shows the displacement of Li ions participating in the second cooperative event (gray shading in panels (**a**), (**b**)).

an approximate window $t = 5-9$ ps, and involves the displacement of five distinct Li ions. This event can be approximately divided into an initial three-ion event that is followed soon afterward by a four-ion process. The ions participating in the four-ion event are highlighted in blue in Fig. 3a. (The preceding three-ion event also involves two of these ions; the third participating ion is shown without shading in Fig. 3a.) The displacements of the ions participating in the four-body event (relative to their positions at the start of the first residence time) are shown in Fig. 3c. These data show that two of the ions undergo large displacements of 4–5 Å, while the displacements for the remaining two ions are on the order of 2 Å.

A second cooperative migration event involving two ions occurs at $t = 57$ ps, and is also shown in Fig. 3b. These ions are identified with gray shading in Fig. 3a, and their correlated displacements are shown in the inset of Fig. 3b. Supplementary Figs. 4 and 5 plot summed $h_i$ values as a function of the parameters used in their evaluation, Eq. (3). These data indicate that the same qualitative behavior persists—i.e., the existence of simultaneous migration events—across the range of explored parameters. This suggests that the observed behavior is not an artifact of the sampling procedure. Supplementary Fig. 6 illustrates that at higher temperatures ($T = 500$ and 700 K), cooperative migration events occur with greater frequency and involve larger clusters of Li ions.

Analysis of the MD trajectories by Eq. (3) implies that Li-ion migration events in LPS glass are correlated in time. Are these events also correlated in space? Figure 4 addresses this question by illustrating the real-space displacements associated with the four-ion migration event occurring at $t = 7$ ps in Fig. 3b over a time window corresponding to $t = 0-10$ ps.

Figure 4a illustrates a portion of the computational cell during this migration event, identifying the beginning ("B") and end ("E") positions of the mobile Li ions, and the trajectories connecting them. Similarly, the displacements of nearest-neighbor $PS_4$ anions (labeled numerically) are depicted by plotting their positions at the beginning (translucent) and end (opaque) of the migration interval. An alternative view of this migration event is shown in Supplementary Fig. 7. On the timescale shown here, four adjacent lithium ions undergo migration. Three of these ions, colored orange, teal, and maroon, undergo displacements that impinge upon the previous locations of the others. This behavior indicates that Li migration events are correlated in space (i.e., involve adjacent ions), and is reminiscent of the "string-like" displacements observed in earlier simulations of Lennard–Jonesian glasses[18]. The observation of concerted hopping events in LPS glass is also consistent with earlier models,

which proposed that ion migration in glasses could involve multiple ions migrating simultaneously[78]. Different from those existing models, the present study demonstrates that large reorientations of the neighboring counter ions—in the present case $PS_4$ anions—can also occur during these events (see below). The ion shown in blue depicts the displacement of the fourth, neighboring Li ion that shares anion 11 with one of the other mobile ions.

Importantly, the nearest-neighbor $PS_4$ anions, which comprise the solvating "cage" surrounding the Li ions prior to their migration, also undergo significant rotations during Li migration. Figure 4b illustrates these anion reorientations and displacements relative to their position at $t = 0$ (i.e., the same time origin as in Fig. 3b). Rotational displacements are defined with respect to a vector whose tail is located at the center of mass of the anion, corresponding approximately to the position of the central P ion, and whose head sits on a vertex of the $PS_4$ tetrahedron. (Four of these vectors can be defined; the rotational displacements of these vectors are shown in Fig. 4b.) Linear displacements were tracked based on the position of the central P ion. The shaded yellow region in Fig. 4b represents the time window for the trajectories shown in Fig. 4a.

Regarding rotational displacements, in the case of anion 1, a large rotation of 45° is observed during $t = 5-10$ ps; the anion then remained in that orientation for the following 10 ps. After undergoing a reorientation during $t = 5-7$ ps, anion 3 experiences oscillatory rotations with displacements as large as 30°. In contrast, anion 8 returns to its initial orientation after a rotational displacement that persists from $t = 2-10$ ps. Anions 11 and 18 undergo a sustained reorientation of 41° and 25°, respectively. Last, anion 10 exhibits oscillatory rotations on the order of 10°.

A more detailed view of the migration mechanism is illustrated in Fig. 4c, d. These panels highlight how the anions that are nearest neighbors to the orange and blue Li ions (from Fig. 4a) rotate and/or shift during Li migration. In both of the illustrated migration processes, the mobile Li maintains its coordination with at least one $PS_4$ anion throughout the entire process (e.g., in Fig. 4c, both anions 1 and 3 remain coordinated to Li). Coordination is maintained via rotational displacements of the anions, consistent with a paddlewheel-type mechanism[47,48]. Another feature common to these migration events is the disassociation or "undocking" of Li from a subset of the neighboring anions near the beginning (time "B") of the process, and the subsequent association or "docking" to new anions near the end (time "E").

Additional data in support of low-temperature paddlewheel dynamics are presented in Supplementary Figs. 8–16. These data

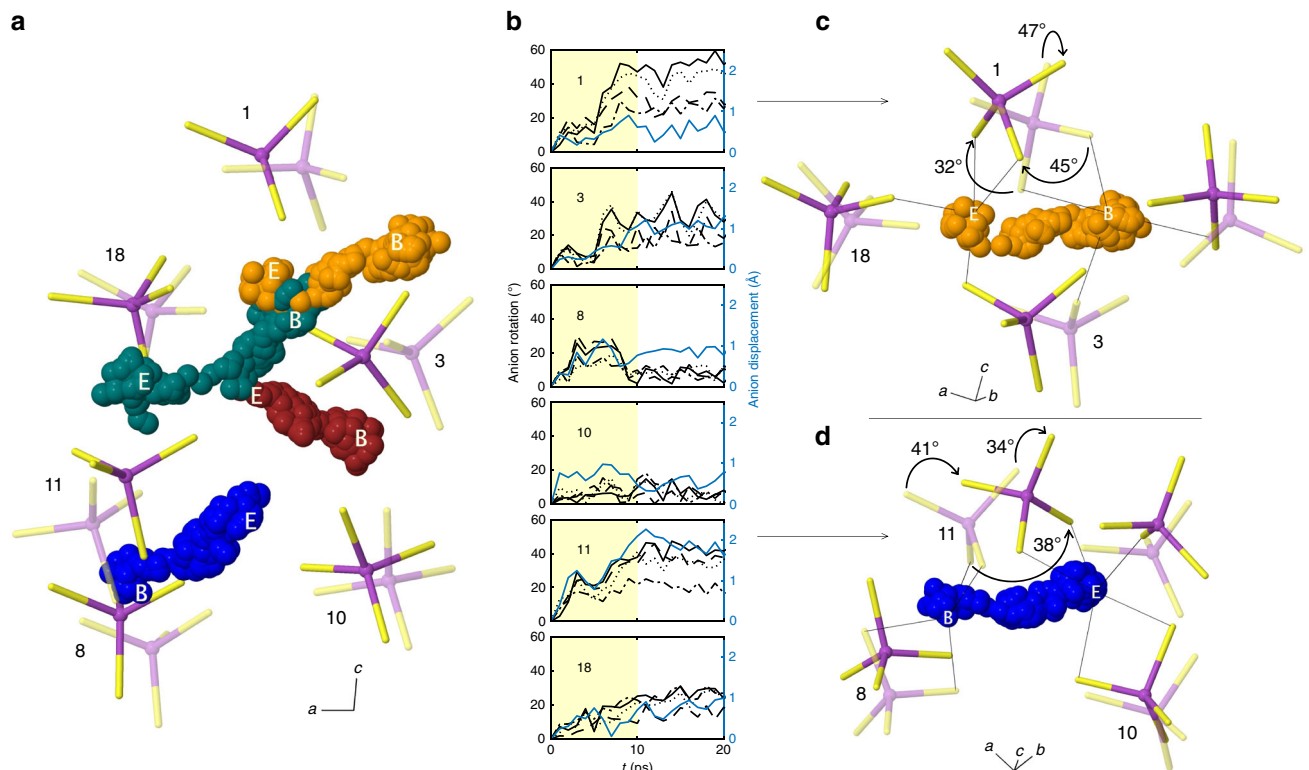

**Fig. 4 Illustration of cation–anion cooperative motion at 300 K. a** Distinct colored spheres represent the positions of four different lithium ions superimposed at 40-fs intervals over a 10-ps trajectory; the initial and final positions of these ions are labeled "B" and "E", respectively. Tetrahedral PS$_4$ anions are colored magenta (phosphorus) and yellow (sulfur). The initial positions of the anions at the start of the migration event are shown with partial transparency; opaque depictions indicate final positions. Numeric labels identify individual anions. For clarity, only a portion of the simulation cell is shown. **b** Angular (black) and linear (blue) displacements of the PS$_4$ anions as a function of simulation time. (Angular displacements are plotted for each of the four vectors parallel to a P–S bond in a PS$_4$ anion.) Yellow shading represents the time window over which a cooperative displacement occurs. **c, d** Diffusion mechanisms for the orange and blue Li ions from panel (**a**), illustrating the coupling of cation transport with the reorientation of anions. Black lines illustrate the evolution of the coordination environment of lithium as it moves from the beginning (B) and end points (E) of the migration displacement. Arrows identify the two anions (numbers 1 and 11) that exhibit the largest rotational displacements. The magnitudes of the largest anion reorientations are identified.

were extracted from 640 ps of simulation time, 480 ps of which were performed at 300 K, with the remaining 160 ps at 400 K. A total of 31 Li migration events were observed across these simulations. Slightly more than half—16 of 31 events—are cooperative processes involving two or more Li ions; six events involve three or more Li. In essentially all migration events a strong correlation exists between displacements of Li and rotational displacements of nearest-neighbor PS$_4$ anions. These correlations manifest at similar times for the occurrence of displacements of Li and PS$_4$, and as a common shape in the displacement plots [see panel (c) in Supplementary Figs. 8–16]. More than three-quarters of the migration events exhibited large rotational displacements of at least 20° in the neighboring anions. The largest rotations occurred during migration events that involved multiple Li ions, with displacements as large as 75° observed in one case (Supplementary Fig. 15). Nevertheless, several single-Li-ion events also coincide with PS$_4$ rotations as large as 45–50° (e.g., see event 3 in Supplementary Fig. 15).

It is instructive to highlight a few examples of paddlewheel dynamics among the low-temperature migration events shown in Supplementary Figs. 8–16. First, event 2 in Supplementary Figs. 9 and 10 is a cooperative event that involves four Li ions, as well as a large displacement of a single sulfur atom (the S atom is identified as atom #116). The displacement profile of Li #1 closely mirrors the rotational displacement of anion #9, with the latter anion rotating 57° over the duration of the event. A plot of the

trajectory of this event is shown in Supplementary Fig. 10. As a second example, Supplementary Fig. 11 shows an "out-and-back" event ("Event 1 + 2") wherein a pair of Li ions (#7 and #32) undergo sequential hops; the first hop out of the initial basin is followed by a subsequent hop back of both ions to the starting basin. Although the rotational displacements associated with this process are smaller (22°), than in other events, a clear correlation exists in the shapes of the anion (rotational) and Li-ion displacements. In this event, anion rotation appears to precede Li-ion displacement. Finally, event 1 in Supplementary Fig. 9 is an example of a single-Li-ion event where the anion rotations are relatively small (<13°), yet track the Li-ion displacements very closely. In this case, anion rotation also precedes Li-ion migration.

Do anion rotations initiate Li migration? Analysis of the 31 low-temperature migration events shown in Supplementary Figs. 8–16 indicates that anion rotation precedes Li migration in more than 60% of the events (19 of 31 events). With one exception (event 5 in Supplementary Fig. 15), the remaining events exhibit displacements that appear to be simultaneous. Establishing causality in these migration events—did an anion rotation initiate a Li displacement, or vice versa?—is not always straightforward, especially for events in which multiple Li and/or anions participate. Nevertheless, based on these data, it is reasonable to conclude that Li displacements and anion rotations are strongly correlated in space and in time.

In addition to quantifying anion reorientations, Fig. 4b and Supplementary Figs. 17–23 illustrate the extent of anion displacements, which were assessed based on the positions of the anions' center of mass (i.e., the position of the central P atom). Supplementary Figure 17 compares high-temperature mean-squared displacement data for the anions and Li ions. These data show that the anions undergo displacements that are much smaller than those of the Li ions. Furthermore, displacement statistics at 300 K were collected for the three P ions nearest neighbor to all migrating Li ions, as shown in Supplementary Figs. 18–23. These data represent an upper bound on anion displacements, as the largest of these displacements tend to occur in the immediate vicinity of a migrating Li. (Anions far from a migrating Li are more stationary.) In all but one case (anion 11 in Fig. 4), these displacements are much less than 2 A, with an average displacement over all observed Li-ion migration events of 0.39 A. For comparison, the average displacement of the migrating Li ions is 2.96 A. Thus, at low T, the displacements of the anions are also much smaller than those of Li. Additional inspection of Supplementary Figs. 18–23 reveals that instances of large anion rotations do not always coincide with significant anion displacements.

To put the rotational behavior of the $PS_4$ anions in glassy LPS in context, similar AIMD calculations on the stable crystalline polymorph at ambient conditions, γ-$Li_3PS_4$, were performed. The rotational and translational behavior in both systems is compared in Supplementary Figs. 24 and 25. In contrast to the large rotations observed for the $PS_4$ tetrahedra in glassy LPS, no significant rotations were observed in the crystalline system over an 80-ps time window. The absence of paddlewheel dynamics in γ-$Li_3PS_4$ correlates with its lower room-temperature conductivity $(3 \times 10^{-7}\, S\, cm^{-1})$[72], which is approximately three orders of magnitude smaller than that of glassy LPS[7,71]. Additional differences regarding rotational displacements in the glass and crystalline phases are illustrated in Supplementary Fig. 26, using the reorientation time–correlation function[74]. These correlations remain high in the crystalline system throughout the simulation, indicating that rotational motion consists primarily of thermal librations. In contrast, the correlations decay rapidly in the glassy system, implying that the $PS_4$ tetrahedra there undergo more substantial rotational displacements. Nevertheless, the rate of decorrelation observed for the glass is slower than that observed in crystalline $Li_2B_{12}H_{12}$[54], suggesting that the rotational dynamics of the $B_{12}H_{12}^{2-}$ anions are more pronounced than for the $PS_4$ tetrahedra of the glass. Supplementary Fig. 27 shows that the rotational motion of the $PS_4$ anions in the glass persists even at higher densities such as $1.76\, g\, cm^{-3}$ (resulting from AIMD simulations conducted at 1 GPa).

In addition, Supplementary Fig. 28 shows that freezing the anion degrees of freedom (i.e., fixed positions with no rotations allowed) dramatically reduces both the number of migration events and their cooperative nature (most events now involve a single Li ion only) at both 500 and 700 K. This behavior should be compared to that of the unconstrained system in Supplementary Fig. 6, where dozens of events occur, with many of these events involving multiple Li ions. A similar constrained calculation at 300 K exhibited no migration events whatsoever. These data provide further evidence that the anion degrees of freedom have a significant impact on the mobility of Li ions.

The above discussion illustrates that the migration of Li cations is correlated with rotational displacements of the $PS_4$ tetrahedra both in space ($PS_4$ adjacent to Li undergoe rotations) and in time (the rotations occur during migration events). It is now demonstrated that the vibrational properties and energetics of the cations and anions are also coupled.

To characterize the vibrational properties, the power spectrum was calculated from the normalized velocity autocorrelation function, defined as[74]

$$\hat{Z}(f) = \int_{-\infty}^{\infty} \frac{\langle x(t')x(t'+t) \rangle}{\langle x^2 \rangle} e^{i2\pi ft}\, dt, \qquad (4)$$

where $x$ represents either the cations' (linear) velocity or the anions' angular velocity. Figure 5 compares the power spectra for LPS glass to those of crystalline γ-$Li_3PS_4$ at 300 K[72]. Several qualitative features are noteworthy. First, these data show that the peak in the lithium vibrational spectrum in the glass is broader and shifted to lower frequencies (~6 THz) compared with that in the crystal (~10 THz). In addition, the glass has a higher density of Li modes in the low-frequency region. Muy et al.[79] have shown that a common feature of fast lithium conductors is low vibration frequencies. These lower frequencies are characteristic of larger thermal displacements, which correlate with a greater probability for migration. Similarly, the anion spectra in the glass also exhibit a greater density of libration modes at lower frequency. The softening of both the Li and anion modes in the glass may reflect the relatively lower density of the amorphous phase, and its associated shallower potential energy surface. Although the glass and crystal anion spectra are similar for frequencies greater than 5 THz, the glass spectrum exhibits a low-frequency peak near 2.5 THz that is absent (or suppressed) in the crystal. A peak at nearly the same frequency has been reported for collective angular motion of $B_{12}H_{12}^{2-}$ anions in *closo*-borate SEs[54]. Finally, the spectra for the glass and the crystal exhibit differing degrees of overlap between the Li vibrational and anion librational modes. In the glass, there is a large overlap between the Li- and the anion spectrum. In contrast, these modes are more separated in the crystal. This behavior suggests that in the glass, Li is more strongly coupled to the rotational motion of the anions. Consequently, anharmonic thermal librations of the anions in the glass have the potential to transfer momentum to the cations, thereby creating a driving force for Li migration, consistent with a paddlewheel-like effect[54]. A similar coupling of cation–anion dynamics was reported for $Li_2B_{12}H_{12}$[54].

Regarding energetics, the energy barriers associated with $PS_4$ rotation and Li migration were extracted from the slopes of Arrhenius plots of the (translational) diffusivity of Li and the rotational diffusivity of the anions. These diffusivity coefficients

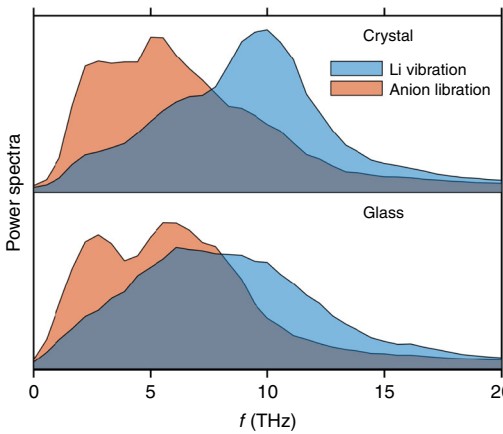

**Fig. 5 Power spectra of the normalized velocity autocorrelation function for crystalline γ-$Li_3PS_4$ (top) and glassy (bottom) $Li_3PS_4$ at 300 K.** Lithium vibrations are shown in blue, and anion librations appear in red.

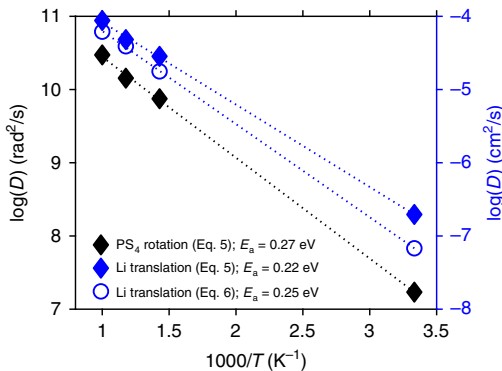

**Fig. 6 Calculated Arrhenius plots and activation energies for anion rotational diffusion and Li translational diffusion in 75Li$_2$S–25P$_2$S$_5$ glass.** Data from the Green–Kubo formula (Eq. 5) are shown with diamonds; data from the Einstein formula (Eq. 6) appear as open blue circles. The dotted lines are a linear fit to the data at 1000, 850, and 700 K. Values at 300 K are extrapolated from the high-temperature data.

were evaluated using the Green–Kubo formula[74]

$$D = \frac{1}{3N} \sum_i^N \int_0^\infty \langle x_i(t') x_i(t'+t) \rangle dt, \tag{5}$$

and are shown as blue and black diamonds in Fig. 6. For comparison, the translational diffusion coefficient for Li was also evaluated with the Einstein formula

$$D = \lim_{t \to \infty} \frac{1}{6tN\tau} \sum_i^N \int_0^\tau |r_i(t'+t) - r_i(t')|^2 dt', \tag{6}$$

and is shown in Fig. 6 as open blue circles. As expected, for lithium, translational diffusion of both formulas yields comparable values for $D$, and for the activation energy, 0.22 eV (Eq. (5)) and 0.25 eV (Eq. (6)). [The extrapolated diffusivities at 300 K are $2.0 \times 10^{-7}$ cm$^2$ s$^{-1}$ (Eq. (5)) and $6.8 \times 10^{-8}$ cm$^2$ s$^{-1}$ (Eq. (6)); the predicted conductivities from the Nernst–Einstein equation are $19 \times 10^{-3}$ S cm$^{-1}$ and $7.0 \times 10^{-3}$ S cm$^{-1}$, respectively.] The measured conductivity and activation energy of conduction for 75Li$_2$S–25P$_2$S$_5$ glass reported by Dietrich et al. are $2.8 \times 10^{-4}$ S cm$^{-1}$ and 0.38 eV, respectively[71]. Ohara et al. reported a similar experimental conductivity of $3.0 \times 10^{-4}$ S cm$^{-1}$ [7], while Hayashi et al.[80] reported an activation energy of 0.35 eV. The slightly lower conductivities (and larger activation energies) reported by experiments may arise from the presence of small quantities of P$_2$S$_6^{4-}$ and/or P$_2$S$_7^{4-}$ anions, or from unreacted Li$_2$S, in those samples. These species are generally thought to suppress Li-ion conductivity[7,71].

Importantly, the activation energy for anion rotation is predicted to be 0.27 eV, which is comparable to, but slightly larger than that predicted for lithium translation, 0.22 eV, using the same formalism (Eq. (5)). The observed activation energies in LPS glass mimic those of the sulfates, where the rotational activation energy for SO$_4$ tetrahedra, 0.40 eV, was found to be similar to, but slightly larger than that for lithium migration, 0.34 eV[45]. The energetic similarity of these barriers in LPS glass provide additional evidence that a strong dynamical coupling between anions and cations underlies the facile transport of Li in this material.

Supplementary Fig. 29 compares the mean square displacement and diffusion coefficient for Li ions with, and without, anion constraints. Freezing the anion degrees of freedom reduces the diffusion coefficient by approximately two orders of magnitude at 300 K, and increases the activation energy by 0.1 eV.

Based on a combination of the reorientation autocorrelation vs. volume data, Supplementary Fig. 26, and the frozen anion simulations, Supplementary Fig. 29—the latter showing a moderate increase in activation energy upon freezing—anion rotations are best described as a contributing factor, but not the sole factor, in facilitating Li-ion mobility in the glass. Stated differently, high cation mobility in the glass appears to result from a combination of paddlewheel dynamics similar to those observed in high-temperature crystalline conductors (e.g., Li$_2$SO$_4$[47] and *closo*-boranes[53]), and from features commonly associated with transport in glasses, i.e., occupancy of higher-energy sites/shallower potential wells[78]. Combining the observations of long decay times for anion reorientation at 300 K (Supplementary Fig. 26) with the presence of occasional rotational displacements of the anions during Li-ion migration events (Supplementary Figs. 8–16), anion dynamics in the glass at 300 K are best described as reorientations rather than as free rotations.

**Explanation for paddlewheel dynamics.** Based on the analyses described above, we propose that the 75Li$_2$S–25P$_2$S$_5$ glass exhibits three features that enable paddlewheel dynamics at low temperature. First, this glass contains PS$_4$ complex anions; rotations of these anions will exert a force on the cations. Second, the glass has a lower density than the crystalline analog, γ-Li$_3$PS$_4$. This lower density provides additional free volume for PS$_4$ rotations[47,53]. This behavior is demonstrated in Supplementary Fig. 26, which displays the reorientation autocorrelation function for the anions. These data show that the orientations of the PS$_4$ anions become increasingly disordered as the density decreases. In contrast, our calculations on γ-Li$_3$PS$_4$ show that PS$_4$ rotations are suppressed in this crystalline phase at low temperature, Supplementary Figs. 24 and 26. Supplementary Figure 26a also shows that anion reorientations in the glass are more prevalent than in γ-Li$_3$PS$_4$, even in cases where the densities of these two systems are similar. This suggests that the amorphous structure of the glass, and the associated shallower potential energy surface, also facilitates anion reorientations.

Third, while a lower density (relative to the crystalline phase) is a property common to any glass, a potentially distinguishing feature of the 75Li$_2$S–25P$_2$S$_5$ composition is the absence of a long-range covalent network in its atomic structure. To understand the impact of this network, it is instructive to compare the structure of the present glass, which contains independent PS$_4$ tetrahedra, to the structure of silica glass[81]. In the latter case, the SiO$_4$ tetrahedra that comprise the glass's building blocks are linked via an irregular, covalently bonded rigid network formed by bridging oxygen. In the case of the LPS glass, such a network could be formed by connecting PS$_4$ tetrahedra through bridging S, or from longer-chain P$_x$S$_y$ anions[71]. The presence of these networks and/or longer-chain P$_x$S$_y$ impedes Li migration[71]. This behavior may arise from the P$_x$S$_y$ network former interfering with the rotational behavior of the smaller PS$_4$ anions. Notably, the 75–25 glass composition has been shown to maximize the number of PS$_4$ anions, while minimizing longer-chain P$_x$S$_y$ components[71]. While the absence of a covalent network and/or long-chain anions is clearly an important feature of the present glass, this behavior could in principle be replicated in other glass compositions by selecting the glass modifier or salt additive so as to ensure complete breakup of the anionic network into independent, compact—and presumably rotatable—complex anions, such as tetrahedra. Microscopically, this could be accomplished by converting all bridging species into their non-bridging analogs. In the $Q_n$ notation of ref. [82], such a scenario corresponds to the case where $n = 0$.

## Discussion

The search for SEs having high ionic conductivities will be aided by an understanding of the atomic-scale mechanisms that foster ionic transport. In the case of crystalline solids, due to the periodic nature of the crystal structure, these mechanisms are becoming increasingly well understood. Consequently, design guidelines for crystalline conductors are starting to emerge[83,84].

In the case of amorphous materials, such as glasses, this understanding is far less developed. This is due to complicating factors, such as the existence of a distribution of site energies[85] and activation barriers for the migrating ion[86,87]. These features can result in deviations from Arrhenius behavior in the conductivity[88]. Hence, rules for achieving high ionic conductivity in this class of materials remain poorly defined. The present study aims to narrow this knowledge gap by reporting ab initio molecular dynamics simulations on the prototype Li-ion-conducting glass, $75Li_2S–25P_2S_5$. A computational model of the amorphous structure was developed and shown to reproduce the short-range order reported in recent neutron measurements.

A noteworthy aspect of the present study is the observation of the detailed mechanism for lithium migration. Li was observed to migrate at room temperature via a concerted process involving both multiple Li ions and dynamic coupling to the rotational motion of the $PS_4^{3-}$ tetrahedra. This latter effect, commonly referred to as the "paddlewheel" mechanism, has historically been reported primarily in crystalline phases that are stabilized at elevated temperatures. Unlike those systems, paddlewheel dynamics are present in the glass at low temperatures.

Paddlewheel contributions to Li mobility were demonstrated by analyzing spatial, temporal, vibrational, and energetic correlations. First, during their migration, Li ions were observed to remain coordinated to a subset of their neighboring $PS_4^{3-}$ tetrahedra via simultaneous, quasi-permanent reorientations of these anions, consistent with the paddlewheel effect. Second, the power spectra of the velocity autocorrelation function exhibit a strong overlap between the lithium vibrational and anion librational modes. This suggests significant "cross-talk" between the vibrations of Li and the anions. Third, the activation energies for anion rotation (0.27 eV) and lithium translation (0.22–0.25 eV) were found to be similar. Fourth, the anion dynamics in the glass were shown to differ markedly from those in the crystalline ($\gamma$-$Li_3PS_4$) analog. In the latter system, contributions from anion reorientations are negligible, and the conductivity is significantly reduced.

In total, these data suggest that glasses based on complex anions, and whose atomic structure does not exhibit a long-range covalent network, may be fertile ground in the search for new SEs. These features are expected to enhance cation mobility at low temperatures by fostering paddlewheel dynamics.

## Methods

First-principles calculations were performed with the Vienna ab initio simulation package (VASP)[89–91]. Core–valence electron interactions were treated with the projector-augmented wave method[92,93]. The generalized gradient approximation in the formulation of Perdew–Burke–Ernzerhof was used for exchange and correlation[94]. The computational cell consisted of lithium and *ortho*-thiophosphate ions ($PS_4^{3-}$). A Monte Carlo annealing procedure employing a classical interatomic potential was used to generate the initial disordered atomic structure (Amorphous Cell Module and the COMPASS II potential from Materials Studio)[95,96] comprising 60 lithium ($Li^+$) and 20 $PS_4^{3-}$ ions at an initial density of $1.9\ g\ cm^{-3}$ in a computational cell having periodic boundary conditions. Starting from the Monte Carlo-generated structure, DFT geometry optimization calculations followed by 29 ps (ps) of ab initio melt-and-quench MD were used to generate a plausible structure for glassy $Li_3PS_4$. Parrinello–Rahman dynamics[97,98] with variable cell shape and volume (NPT ensemble) were employed in combination with a Langevin thermostat[99]. A time step of 2 fs was used to integrate the equations of motion. To minimize Pulay stresses, the plane-wave cutoff energy was set to 400 eV. The friction coefficient for the atomic species and lattice was

$10\ ps^{-1}$. The lattice mass was set to 1000 amu. A Gamma-only *k*-point mesh was used. The melt-and-quench MD procedure involved an initial equilibration at 300 K for 3 ps, heating to 1000 K at 70 K $ps^{-1}$, a 3-ps hold at 1000 K, cooling to the desired temperature at the same 70 K $ps^{-1}$ rate, and finally, equilibration for an additional 3 ps at the target temperature. A schematic of this procedure is shown in Supplementary Fig. 1. Following 3 ps of thermal equilibration, MD simulations at 1000, 850, 700, 500, 400, and 300 K for a minimum duration of 80 ps at each temperature were performed. Since the present study emphasizes low-temperature migration mechanisms, multiple independent MD runs were performed at 300 and 400 K: in total, 480 ps of simulation time was amassed at 300 K, with an additional 160 ps at 400 K. In cases where multiple MD runs were performed at the same temperature, distinct initial velocities were assigned to each system in order to de-phase the dynamics.

## Data availability

Calculated structure data and the raw data used to characterize glass properties are available from the corresponding author upon reasonable request.

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

## Acknowledgements

This work was supported as part of the Joint Center for Energy Storage Research, an Energy Innovation Hub funded by the U.S. Department of Energy, Office of Science, and Basic Energy Sciences.

## Author contributions

J.G.S. conducted the computational components of the project. All authors contributed to the drafting of the paper. D.J.S. conceived the project idea.

## Competing interests

The authors declare no competing interests.
