## [Peer Review File · Nature Communications]

Reviewers' comments:

Reviewer #1 (Remarks to the Author):

Coupling of the anion rotation with the lithium translation is examined in details from the MD simulations of sulfide glasses that serve as a prototype for the conductive solid state amorphous (and crystalline) electrolytes. This study is well-performed, careful and clearly articulated. I expect that many researchers in the field will find it inspirational. While reviewer 3 points that oxygen and other anion deformations are often coupled with the cation motion in cathodes and solid electrolytes, making transitions softer and faster, I agree with the point made by authors that these local vibrations or small re-arrangements are quite different from the more pronounced Paddle-Wheel Mechanism discussed in this manuscript. I recommend this manuscript for publication in Nature Communications after authors address my specific comments below.

Was protocol shown in Figure S1 applied to all "6 independent MD runs performed at 300 K". How do authors know that systems were decorrelated and independent?

What is the distance "a" in eq. 3?

In Fig. S7 and Fig. 4, the initial and final positions of the Li⁺ cations are labeled but the initial and final positions of the sulfur of PS4 are not. Could author label S with 1,2,3,4 in Fig. S7 to make it possible to see how the PS4 anions rotated during this simulation from the initial to final positions. This should clearly show that a "new" S comes to the same position after helping a Li⁺ to move via the PS4 rotation. Such diagram will help to differentiate this work from the other work mentioned by the 3d reviewer (Chem. Mater. 2015, 27, 6016–6021) where oxygens deformed but returned to the same positions after assisting cation to overcome a barrier. Rotation of the PS4, OH or B12H12 anions is different assuming it is not a partial rotation with a return to the same position. Figure 4b indicates that it is a combination of rotations that "complete" and "return" to the same position and it varies with density according to Figure S19.

Figure 4 shows that some anions have displacements of 2 Å while the most active Li move 5 Å (Figure 3). Could authors add MSD for anions vs. different temperatures in SI to convenience readers that anions do not move much during MD simulations, especially, at high temperature. If anions move at high temperature, their diffusion coefficient vs. temperature should be included in Figure 6 as it could also be coupled to Li translation (in addition to anion rotation).

Anion translation is also important because freezing anions (Figure S21) freezes both anions translation and rotation. The former is not mentioned. I think that this could be an important depending on the magnitude of the anion MSD at higher temperatures.

Is there a correlation of the anion MSD with the anion rotations shown in Figure S19? If there is, it is important to quantify it.

A cooperative motion of anions with lithium might also account to the lower experimental conductivity compared to the predicted value that is based only on the lithium diffusion without including anions. Again, if anions do not move at all, it is not an issue.

There is a significant overlap with the analysis of paddlewheel reported for (Li/Na)B12H12 solid state

conductor (power spectrum, rotational relaxation, etc) reported in Chem. Mater. 2017, 29, 9142–9153. It could be helpful to specify in the discussion the main differences in the mechanism between sulfide glasses and this system could help readers contrast and better understand both works.

Page 2: "Unfortunately, exploiting these rotational degrees of freedom presents a challenge, as the paddlewheel mechanism is typically observed in high-temperature polymorphs having expanded volumes. 39–43,45–54,61

Note that a paddlewheel effect was also observed for $\text{Li}_2+x\text{OH}(1-x)\text{Cl}$ solid state conductors for cubic phase near to room temperature but called "rotating door" mechanism, see Song Protons Enhance Conductivities in Lithium Halide Hydroxide/Lithium Oxyhalide Solid Electrolytes by Forming Rotating Hydroxy Groups. Adv. Energ. Mater. 2018, 8 (3), 1700971-8, followed by Howard Fundamental aspects of the structural and electrolyte properties of Li_2OHCl from simulations and experiment. Phys. Rev. Mater. 2017, 1 (7), 075406. A clear correlation between the anion rotation and lithium diffusion (space and time) was also shown for these systems, albeit with the DFT-based MD simulations performed at higher temperature but experiments indicate that this mechanism persists down to room temperature as Arrhenius scaling of conductivity was observed but because some of the analysis of the coupling of anion rotation with Li^+ motion is similar, I think that this work should be referenced.

It would be helpful to give in Figure S3b the neutron-weighted partial contributions to $G(r)$ to clearly show which PDFs contribution to each peak. Otherwise, it is not clear by a strong peak for Li-S PDF around 2.5 Å yields a minimum in Figure S3. This will also clarify sensitivity of the neutron weighted $G(r)$ to different contributions.

Oleg Borodin

Reviewer #2 (Remarks to the Author):

The authors presented comprehensive diffusion mechanisms studies in glass Li_3PS_4 via ab initio MD simulations. The authors clearly demonstrated that the previously-proposed "Paddlewheel" mechanism contributes to the Li-ion diffusion at temperatures as low as 300K in the glass phase but not in the crystalline phase. This is not a trivial task, and the authors also did a very carefully job in generating representative amorphous structures. They considered the "Paddlewheel" mechanism is the key to the fast ionic conductivity in glass Li_3PS_4 .

While I applaud for the detailed simulation work on identifying the Li diffusion mechanism in the glass Li_3PS_4 phase. I would suggest the authors could comment on ionic conductivity, not just the diffusivity.

When Li hops from one site to another site, these sites will have different energy levels in the glass structure, like those shown in PCCP 11, 3210, 2009. The authors fit one activation energy for diffusivity, in fact, the amorphous structure has a distribution of the activation energies, as discussed in PRL 109, 075901, 2012. The distribution of the activation energies in amorphous was also used to explain the non-Arrhenius conductivity in Li-P-S glass (Solid State Ionics, 176, 2349, 2005). However, these theories on diffusion in glass structures are not discussed in the current manuscript. To advance the field, I hope the authors could comment on the distribution of site energy, the percentage of low energy migration pathway, and the distribution of activation energies in amorphous.

Reviewer #1 (Remarks to the Author): Coupling of the anion rotation with the lithium translation is examined in details from the MD simulations of sulfide glasses that serve as a prototype for the conductive solid state amorphous (and crystalline) electrolytes. This study is well-performed, careful and clearly articulated. I expect that many researchers in the field will find it inspirational. While reviewer 3 points that oxygen and other anion deformations are often coupled with the cation motion in cathodes and solid electrolytes, making transitions softer and faster, I agree with the point made by authors that these local vibrations or small re-arrangements are quite different from the more pronounced Paddle-Wheel Mechanism discussed in this manuscript. I recommend this manuscript for publication in Nature Communications after authors address my specific comments below.

Response: We thank the referee for their favorable comments, careful reading of our manuscript, and for the suggestions provided below. A point-by-point response follows.

(1) Was protocol shown in Figure S1 applied to all “6 independent MD runs performed at 300 K”. How do authors know that systems were decorrelated and independent?

Response: These MD runs were derived from the same ‘seed’ structure, but were initiated with different velocities and de-phased for 3 ps before statistics were collected. (A similar procedure is used in Voter’s Parallel Replica Dynamics to generate independent MD trajectories from the same starting structure.) Additional evidence that the systems are independent can be seen in Figures S8-S14, which show that the migration events for these systems are quite different – they involve different atoms and occur at different times.

Changes to manuscript: Added text, page 3: “In cases where multiple MD runs were performed at the same temperature, distinct initial velocities were assigned to each system in order to de-phase the dynamics.”

(2) What is the distance “a” in eq. 3?

Response: The hop length, a , is set to 1.6 Å. This is the same value used in Ref. 35. This definition has been added to the manuscript.

Changes to manuscript: Added text, page 4: “The displacement threshold, a , was set to 1.6 Å, in accordance with an earlier study[35]. Figures S4-S5 show the effect of varying the size of this threshold.”

(3) In Fig. S7 and Fig. 4, the initial and final positions of the Li⁺ cations are labeled but the initial and final positions of the sulfur of PS4 are not. Could author label S with 1,2,3,4 in Fig. S7 to make it possible to see how the PS4 anions rotated during this simulation from the initial to final positions. This should clearly show that a “new” S comes to the same position after helping a Li⁺ to move via the PS4 rotation. Such diagram will help to differentiate this work from the other work mentioned by the 3d reviewer (Chem. Mater. 2015, 27, 6016–6021) where oxygens deformed but returned to the same positions after assisting cation to overcome a barrier. Rotation of the PS4, OH or B12H12 anions is different assuming it is not a partial rotation with a return to the same position. Figure 4b indicates that it is a combination of rotations that “complete” and “return” to the same position and it varies with density according to Figure S19.

Response: This suggestion has been implemented.

Changes to manuscript: Sulfur labels have been added to Figure S7. In Fig. 4, S atoms that undergo large rotations are identified. These demarcations have been described in the respective figure captions.

(4) Figure 4 shows that some anions have displacements of 2 Å while the most active Li move 5 Å (Figure 3). Could authors add MSD for anions vs. different temperatures in SI to convenience readers that anions do not move much during MD simulations, especially, at high temperature. If anions move at high temperature, their diffusion coefficient vs. temperature should be included in Figure 6 as it could also be coupled to Li translation (in addition to anion rotation). Anion translation is also important because freezing anions (Figure S21) freezes both anions translation and rotation. The former is not mentioned. I think that this could be an important depending on the magnitude of the anion MSD at higher temperatures.

Response: We have followed the referee’s suggestion and analyzed the anions’ displacements. Overall, these displacements are much smaller than those of Li.

Changes to manuscript: Added anion MSD and low-T displacements during cation hopping events to SI as Figures S17 and S18-S23, respectively. Added text to manuscript (page 6): “In addition to quantifying anion reorientations, Figures 4b and S17-S23 illustrate the extent of anion displacements, which were assessed based on the positions of the anions’ center of mass (i.e., the position of the central P atom). Figure S17 compares high-

temperature mean squared displacement data for the anions and Li-ions. These data show that the anions undergo displacements that are much smaller than those of the Li-ions. Furthermore, displacement statistics at 300 K were collected for the 3 P ions nearest-neighbor to all migrating Li ions, as shown in Figs. S18-S23. These data represent an upper bound on anion displacements, as the largest of these displacements tend to occur in the immediate vicinity of a migrating Li. (Anions far from a migrating Li are more stationary.) In all but one case (anion 11 in Fig. 4) these displacements are much less than $2A$, with an average displacement over all observed Li-ion migration events of $0.39 A$. For comparison, the average displacement of the migrating Li ions is $2.96 A$. Thus, at low-T the displacements of the anions are also much smaller than that of Li.”

(5) Is there a correlation of the anion MSD with the anion rotations shown in Figure S19? If there is, it is important to quantify it.

Response: Since the anion displacements are small (as described above), we have not attempted to quantify such a correlation. Nevertheless, inspection of Figs. S18-S23 reveals that instances of large anion rotations do not always coincide with significant anion displacements.

(6) A cooperative motion of anions with lithium might also account to the lower experimental conductivity compared to the predicted value that is based only on the lithium diffusion without including anions. Again, if anions do not move at all, it is not an issue.

Response: As discussed above, anion displacements were found to be small relative to those of Li, so this does not appear to be an issue.

(7) There is a significant overlap with the analysis of paddlewheel reported for (Li/Na)B₁₂H₁₂ solid state conductor (power spectrum, rotational relaxation, etc) reported in Chem. Mater. 2017, 29, 9142–9153. It could be helpful to specify in the discussion the main differences in the mechanism between sulfide glasses and this system could help readers contrast and better understand both works.

Response: We have adopted this suggestion.

Changes to manuscript: Added text to page 7: (regarding Fig S26) “Nevertheless, the rate of decorrelation observed for the glass is slower than that observed in crystalline Li₂B₁₂H₁₂,⁵⁴ suggesting that the rotational dynamics of the B₁₂H₁₂²⁻ anions in that system are more pronounced than for the PS₄ tetrahedra of the glass.”

(regarding Fig 5.) “A peak at nearly the same frequency has been reported for collective angular motion of B₁₂H₁₂²⁻ anions in closo-borate solid electrolytes.⁵⁴” and “A similar coupling of cation-anion dynamics was reported for Li₂B₁₂H₁₂.⁵⁴”

(8) Page 2: “Unfortunately, exploiting these rotational degrees of freedom presents a challenge, as the paddlewheel mechanism is typically observed in high-temperature polymorphs having expanded volumes. 39–43,45–54,61 Note that a paddlewheel effect was also observed for Li₂+xOH(1-x)Cl solid state conductors for cubic phase near to room temperature but called “rotating door” mechanism, see Song Protons Enhance Conductivities in Lithium Halide Hydroxide/Lithium Oxyhalide Solid Electrolytes by Forming Rotating Hydroxy Groups. Adv. Energ. Mater. 2018, 8 (3), 1700971-8, followed by Howard Fundamental aspects of the structural and electrolyte properties of Li₂OHCl from simulations and experiment. Phys. Rev. Mater. 2017, 1 (7), 075406. A clear correlation between the anion rotation and lithium diffusion (space and time) was also shown for these systems, albeit with the DFT-based MD simulations performed at higher temperature but experiments indicate that this mechanism persists down to room temperature as Arrhenius scaling of conductivity was observed but because some of the analysis of the coupling of anion rotation with Li⁺ motion is similar, I think that this work should be referenced.

Response: References to these papers have been added.

(9) It would be helpful to give in Figure S3b the neutron-weighted partial contributions to G(r) to clearly show which PDFs contribution to each peak. Otherwise, it is not clear by a strong peak for Li-S PDF around $2.5 A$ yields a minimum in Figure S3. This will also clarify sensitivity of the neutron weighted G(r) to different contributions.

Response: This suggestion has been adopted.

Changes to manuscript: Added new Figure S3(b) and associated caption text to Figure S3.

Reviewer #2 (Remarks to the Author): The authors presented comprehensive diffusion mechanisms studies in glass Li₃PS₄ via ab initio MD simulations. The authors clearly demonstrated that the previously-proposed “Paddlewheel” mechanism contributes to the Li-ion diffusion at temperatures as low as 300K in the glass phase but not in the crystalline phase. This is not a trivial task, and the authors also did a very carefully job in generating representative amorphous structures. They considered the “Paddlewheel” mechanism is the key to the fast ionic conductivity in glass Li₃PS₄.

Response: We thank the referee for their favorable comments, careful reading of our manuscript, and for the suggestions provided below. A point-by-point response follows.

While I applaud for the detailed simulation work on identifying the Li diffusion mechanism in the glass Li₃PS₄ phase. I would suggest the authors could comment on ionic conductivity, not just the diffusivity.

Response: Please note that a brief discussion of ionic conductivity already exists on page 8.

When Li hops from one site to another site, these sites will have different energy levels in the glass structure, like those shown in PCCP 11, 3210, 2009. The authors fit one activation energy for diffusivity, in fact, the amorphous structure has a distribution of the activation energies, as discussed in PRL 109, 075901, 2012. The distribution of the activation energies in amorphous was also used to explain the non-Arrhenius conductivity in Li-P-S glass (Solid State Ionics, 176, 2349, 2005). However, these theories on diffusion in glass structures are not discussed in the current manuscript. To advance the field, I hope the authors could comment on the distribution of site energy, the percentage of low energy migration pathway, and the distribution of activation energies in amorphous.

Response: We agree that there remain other interesting phenomena to examine in the LPS system. We have added text to the manuscript that cites the suggested papers and mentions that more work needs to be done.

Unfortunately, many of the additional investigations suggested by the referee are computationally expensive and non-trivial to pursue (such as the distribution of site energies and energy barriers). For example, since the most important low-energy barriers are not known *a priori*, dozens of barriers must be evaluated to achieve a representative sampling of their distribution. These calculations would need to be performed on (the present) relatively large simulation cell with no simplifying symmetry. In fact, our prior study of these same issues was the subject of an entire paper (Chem. Mater., 26, 2952 (2014)!

In addition, we reiterate that the focus of the present study is on the rotational dynamics of the complex anions. Hence, the phenomena mentioned by the referee – while interesting – are orthogonal to our focus. Consequently, we suggest that these additional investigations are best reserved for subsequent publications where they can be treated with the detail they deserve.

Changes to manuscript: added text and references (page 9): “...In the case of amorphous materials, such as glasses, this understanding is far less developed. This is due to complicating factors such as the existence of a distribution of site energies⁹⁶ and activation barriers for the migrating ion.^{97,98} These features can result in deviations from Arrhenius behavior in the conductivity of glasses.⁹⁹”

REVIEWERS' COMMENTS:

Reviewer #1 (Remarks to the Author):

Authors adequately address my comments that further confirmed the claimed mechanism present at both low and high temperatures. I think that this manuscript will be of interest to a wide range of solid state chemists and battery researchers. I recommend it for publication.

Reviewer #2 (Remarks to the Author):

The authors agreed that the simulations are only about the Li-ion "migration" process and they only calculated Li-ion diffusivity, not the Li ionic conductivity. I suggest "Conductivity" should be replaced by "diffusivity" in the following places to avoid exchanging the two concepts, especially their relationship is more complicated for amorphous.

a) In the abstract "Furthermore, the dynamics in the glass are shown to differ from those in the stable crystalline (g-Li3PS4) analogue, where contributions from anion reorientations are negligible and the conductivity is several orders of magnitude smaller." [They only computed D]

b) On page 2, "More specifically, ab initio molecular dynamics (AIMD) simulations were used to reveal the static and dynamic atomic scale features that underlie ionic conductivity in glassy Li3PS4." [They only studied migration or diffusion]

RESPONSE TO REVIEWERS' COMMENTS:

Reviewer #1 (Remarks to the Author): Authors adequately address my comments that further confirmed the claimed mechanism present at both low and high temperatures. I think that this manuscript will be of interest to a wide range of solid state chemists and battery researchers. I recommend it for publication.

Response: We thank the reviewer for these favorable comments.

Reviewer #2 (Remarks to the Author): The authors agreed that the simulations are only about the Li-ion “migration” process and they only calculated Li-ion diffusivity, not the Li ionic conductivity. I suggest “Conductivity” should be replaced by “diffusivity” in the following places to avoid exchanging the two concepts, especially their relationship is more complicated for amorphous.

Response: We thank the reviewer for these favorable comments. Changes to the wording consistent with the Referee’s suggestions have been implemented.

a) In the abstract “Furthermore, the dynamics in the glass are shown to differ from those in the stable crystalline (g-Li3PS4) analogue, where contributions from anion reorientations are negligible and the conductivity is several orders of magnitude smaller.” [They only computed D]

Response: We replaced “the conductivity” with “the rate of ionic transport”

b) On page 2, “More specifically, ab initio molecular dynamics (AIMD) simulations were used to reveal the static and dynamic atomic scale features that underlie ionic conductivity in glassy Li3PS4.” [They only studied migration or diffusion]

Response: We replaced “conductivity” with “diffusivity”